# Enotourism in Southern Spain: The Montilla-Moriles PDO

**DOI:** 10.3390/ijerph19063393

**Published:** 2022-03-13

**Authors:** Jose Antonio Cava Jimenez, Mª Genoveva Millán Vázquez de la Torre, Mª Genoveva Dancausa Millán

**Affiliations:** 1Department of Agriculture Economics, Finance Accounting, Córdoba University, 14071 Córdoba, Spain; jcava@uco.es; 2Department Quantitative Methods, Universidad Loyola Andalucía, 14004 Córdoba, Spain; 3Department of Statistics, Córdoba University, 14071 Córdoba, Spain; z62damim@uco.es

**Keywords:** gastronomic routes, protected designation of origin, gastronomic tourism, logit model, Montilla-Moriles, ARIMA

## Abstract

The profile of tourists during the COVID-19 pandemic is changing toward those seeking health, safety and quality products. One of the modalities that best adapts to these needs is gastronomic tourism and, within this segment, wine tourism (enotourism), which can be enjoyed in many areas across the world. The great diversity of grapes, climates, terrains and winemaking processes gives rise to an enormous variety of wines that ensures that no two wines are alike. The current situation of the tourism market necessitates enhancing the uniqueness of areas that offer differentiated products, helping to position such locations as benchmarks for gastronomic tourism. Gastronomic routes provide a way to unify and benefit rural areas through the recently increased demand of tourists seeking to experience regional foods. In this study, the Montilla-Moriles Wine Route is analyzed with the objective of forecasting the demand (using autoregressive integrate moving average, ARIMA models), establishing a tourist profile and calculating the probability that a wine tourist is satisfied with the visit based on their personal characteristics (logit model). The results obtained indicate a slight increase (3.6%) in wine tourists with a high degree of satisfaction, primarily derived from the gastronomic or catering services of the area, from the number of wineries visited, from the treatment received and from the age of the tourist. Consequently, a high percentage of these tourists recommend the route. By increasing the demand for enotourism in this area and applying the results obtained, marketing initiatives could be established, particularly for wine festivals to improve this tourist segment and generate wealth in that area.

## 1. Introduction

The changes that have taken place in recent years in tourism activity and during the COVID-19 pandemic [1,2,3,4] are creating new destinations that, far from the traditional sun and beach destinations, generate complementary routes for wealth and job creation [5,6]. Thus, along with the already classic inland destinations, other tourism products are emerging that attract certain segments of the population. As a result of this demand, new tourist routes are being created, among which gastronomic routes stand out [7].

Until 2019, Spain was considered the second largest tourist power in the world, attracting more than 82 million foreigners, mainly for sun and beach tourism, generating an economic benefit of more than 92 billion euros, which served as the driving force of the economy and allowed the country to overcome the economic crisis of 2009 by compensating for a balance in payments. Nevertheless, the 2020 health crisis paralyzed this sector when the borders closed. Spain is currently recovering slowly by receiving international tourists again, but the number of tourists has not exceeded 40 million [8]. The tourism sector is changing its profile, seeking to revitalize national tourism, but it has to be modernized and adapted to the changing demands of new tourists, who are more concerned with safety and sustainability and less so with cost. Post-COVID-19 tourists [5,6,9] prefer quality services and pay more for a differentiated product with a quality label [6,10,11]. Within these new quality parameters is gastronomic tourism based on protected designation of origin (PDO) and protected geographical indication (PGI) classifications, which guarantee product quality [12,13]. Such destinations are located in rural environments, which have been in high demand by tourists during the pandemic [14].

Several scientific studies indicate that tourism has the ability to improve the lives of people and drive economic growth [15] not only through directly related activities but also by boosting other economic sectors [16]. Moreover, tourism influences the economic activity of certain areas and territories. Gastronomy has the ability to be the cause and effect, gastronomic tourism [7], with the proviso that while food is one of the main motivations for choosing a tourist destination [17], it is not the only one because today’s tourists travel for multiple reasons [18]. The weight of consumers’ opinions in the choice of the final destination must also not be overlooked [19,20].

New tourist patterns, the search for new experiences, and the availability of free time are motivating the evolution of certain types of tourism, such as enotourism [21,22,23,24].

The current situation of the tourism market makes it necessary to promote the uniqueness of certain areas and unify marketing synergies [25] to offer a differentiated product; this approach will help Spain position itself as a benchmark for gastronomic tourism. The search for natural and healthier environments [26] has made rural areas the preferred destinations for travel by Spaniards after the pandemic [27,28] because recreational activities based on elements of rural culture can safely be carried out [29] in these environments. Within its services and products, this tourism segment includes sports [30], nature appreciation [31], learning about the gastronomic products of the area [32,33,34], e.g., coffee, cheeses, wines, mushrooms, and oils, or simply enjoying the tranquility that the rural environment offers. A wide range of complementary activities throughout the year makes it possible to reduce the seasonality typical of the tourism sector and generate more stable jobs [35].

Gastronomy and food products from the land are resources used by agricultural areas for tourism [36,37,38], as they can be channeled through gastronomic routes, which are a means to unify rural areas. Furthermore, the number of tourists who seek to engage with regional foods has recently increased [39].

This research aims to analyze the probability of an enotourist being satisfied with the wine route toured in the Montilla-Moriles PDO, and to predict the demand for wine tourism in this region and compare it with the pre-pandemic figures.

## 2. PDOs and Gastronomic Routes

Tourism, in all its forms, is a means to solve economic and social issues or challenges within interior regions. Changes in the economic and social roles of the traditional production of food products in rural areas, through a restructuring of the productive structure, offer new job opportunities for the population [40]. Tourism may not be the main source of income in rural areas but can provide supplemental income for local inhabitants [41].

New trends in consumer habits have led to a growing interest in higher-quality products, differentiated and adapted to the new needs of different groups and market segments. Given this increase in the consumption of differentiated products based on quality, one of the most recognized strategies in the agri-food sector to achieve this differentiation is geographical indicators of origin and, in particular, PDOs, which integrate in their definition not only the geographical origin but also the tradition and specialization of producing high-quality products with unique features and the regulation and control of mechanisms for their production [42].

PDOs [43] and PGIs, located mainly in rural areas, constitute the system used in Spain for the recognition of high-quality food products, resulting from the unique and differentiating characteristics related to the geographical environment where the raw materials are produced and the products are manufactured and to the influence of human factors involved [44]; however, they are not sufficient to create a tourist product. As such, it is necessary to create tourist routes, so-called gastronomic or food routes [45]. There can be countless activities that tourists can engage in related to the products identified by these routes; for example, visiting producers at their establishments where tourists are shown the preparation processes and allowed to taste the products. In addition, restaurants will offer traditional dishes prepared with products from the area. However, to constitute a gastronomic route (Figure 1), a distinctive product is first needed; that is, a raw material such as wine or oil endorsed by a PDO. Second, an itinerary is needed based on a road network that includes several businesses affiliated with the route such as wineries, restaurants, hotels or shops where the product is marketed or showcased in culinary dishes or accompanying meals. Furthermore, all committed establishments on gastronomic routes must meet certain standards of quality that identify the route. Signage for the route on the road network and to identify the participating establishments along the route must also be provided, and an organization or association must coordinate the different elements of and information about the route.

In Spain, according to the latest data from November 2021, 202 PDOs and 160 PGIs were registered (Table 1). Of these, 96 (46.5%) are the designation of origin of wines, and 43 are PGIs (26.9%). Cheeses, oils, fruit and vegetable products compose the remaining 53.7%. The economic impact of the agri-food sector, according to the Ministry of Agriculture, Fisheries and Food (MAPA) [46], is approximately 5000 million euros. Of this economic value, wines, accounting for approximately 3000 million euros, rank first, followed by spirits, accounting for 450 million euros, followed distantly by cheeses, meat products, turrones (nougats), fruits and virgin olive oils.

In Andalusia, a region located in southern Spain (and the object of this research), 30 designations of origin and 31 geographical indications are registered. Andalusia is the autonomous community with the most designations of origin and PGIs, followed by Castilla y León with 27 and Castilla–La Mancha with 24. By type of product, virgin olive oil stands out, with 13 designations of origin, accounting for 41.9% of the Spanish designations of origin for this product [47]. The wine sector is also very prominent, with eight PDOs. Regarding other wine products, i.e., vinegars, there are only two PDOs recognized throughout all of Spain; for fruits and vegetables, there are four PDOs, and Serrano ham has one PDO, indicating that the autonomous community of Andalusia accounts for a considerable total registered in Spain, i.e., 67% in both cases.

According to Blanco and Riveros [48], a gastronomic route in rural environments associated with a PDO is a form of rural tourism that stimulates new economic activities to maintain and improve the living conditions of the rural population. Such tourism aims to achieve a product that integrates the greatest number of actors, generates more jobs in these areas, and that diversifies the existing offerings [49]. Tourism in these rural areas is not the main source of income and does not saturate the environment but rather contributes supplemental income to the local inhabitants.

Gastronomic routes must be thought of as a tour that is organized in such a way to allow tourists to recognize and enjoy the agricultural and industrial production processes and to taste regional cuisine, expressions of the cultural identity of the region. The routes are organized around a key characteristic product, which often provides the name for the route. In addition, routes offer a series of pleasurable experiences and activities related to distinctive elements; they are organized to consolidate productive regional culture and to enhance regional products to stimulate regional economies through the promotion of products and gastronomic culture [50].

Barrera [51] classifies gastronomic routes as follows:Gastronomic routes by product, which are routes organized on the basis of a specific product, e.g., cheese, oil, and wine;Gastronomic routes by dish, which are organized around the most important prepared dishes; Ethnic-gastronomic routes, which are ventures based on the culinary traditions of immigrant peoples.

In addition, tourists are offered a series of pleasures and activities related to the distinctive elements of products, including visiting cultivation fields and processing plants, learning the history of the evolution of the product, tastings, etc. They are organized to consolidate the productive regional culture, to enhance regional products to stimulate regional economies through the promotion of products and gastronomic culture.

The promotion of food brands through routes, in addition to the place of origin of the product, is a means of promoting typical products of the region, providing added value to the service/product offered to tourist consumers. The promotion of gastronomic and culinary heritage includes not only consumption on the premises but also the acquisition of regional food products as souvenirs, thus increasing the income obtained from native products of the area and making it possible to position the food product in the market.

## 3. Enotourism: The Montilla-Moriles Route

The definition and conceptualization of enotourism are not uniform because they can be analyzed from different perspectives, such as marketing or the motivation of travelers. Following Getz and Brown [52], enotourism is simultaneously a consumer behavior, a strategy to develop a geographical area and the wine market in that area and an opportunity to promote the sale of products by wineries directly to consumers. Recent studies on the subject of wine tourism suggest and promote the idea that food and wine can be, and often are, the primary reason to travel to a certain region and not necessarily a secondary activity of the trip [53].

New tourist patterns, the search for new experiences, and the availability of free time are generating the evolution of certain types of tourism, such as enotourism [21,22,54]. The importance that enotourism has acquired in recent years in different parts of the world has been sufficiently documented [55], for example, Chile [56,57,58]; Hungary [59]; New Zealand [60,61,62,63,64]; South Africa [65,66,67]; Italy [23,68,69]; France [70,71]; the US [55,72,73,74,75]; Portugal [76]; and Spain [18,41,77,78,79]. Such an expansion of tourist destinations brings with it new regions that will boost the economy of these areas. Specifically, in Spain, the Gilbert study [80], which analyzed the importance of wine tourism on the area of La Rioja at the beginning of the nineties of the last century, inspired similar studies in other wine regions, for example, Valencia [81,82,83], Priorato [84], Montilla-Moriles [85,86,87] and Malaga [88,89]. These studies discussed how the development of this tourism product, managed by small and medium enterprises (many of them cooperatives), can serve as a complement to other activities in rural areas, generating wealth and creating jobs. Elias [87] describes enotourism as trips and stays geared toward knowledge of the landscapes, tasks and spaces of winemaking and the activities that increase knowledge and acquisition and generate development in various wine regions.

However, for enotourism to develop effectively, it must be supported by quality products, i.e., PDOs and PGIs, that are regulated by councils to ensure the quality of the products under their umbrella.

In Europe, there are 1322 designations of origin of wine and 388 PGIs (Table 2), with France having the most quality labels (554), followed by Italy (547) and Spain (139). Spain has about one million cultivated hectares. It has the largest area of vineyards in the world, but in terms of production, it ranks third, behind Italy and France. It is also the world leader in the export of wine, exporting more than 22 million hectoliters in 2019, with countries of the European Union being the main consumers.

They are, therefore, well-known Spanish wines in the European Common Market, and consumers of such wines could be potential clients for enotourism, with destinations in any of the 17 autonomous communities of the country because the PDOs are distributed throughout the entire Spanish territory (Figure 2).

However, to create a quality product related to wine, it is necessary to have not only PDOs and PGIs but also routes associated with those products [75]. In Spain, the region with the most wine routes is Castilla León with seven, while Andalusia only has three (Figure 3).

Once a tourist route has been created, the next step is to promote the tourism product [90] for the purpose of reaching the potential demand of wine tourists. Thus, a key question arises: Who are the wine tourists? There are different studies that analyze the characteristics of wine tourists. Among such studies, Charters and Ali-Knight [91] group tourists into four different types:⮚Wine lover. These individuals have a vast education in oenological aspects, and the main reason for their trip is to taste different types of wine, to buy bottles of wine and to learn in situ. They are also very interested in local gastronomy.⮚The connoisseur. These individuals, although they do not have a vast education in oenological issues, know the world of wine relatively well. They usually have a university education, and the main reason for their trip is to put into practice what they have read in different specialized magazines.⮚Wine interested. These individuals do not have technical training in oenological issues but are interested in the world of wine. Visiting wineries is not the main reason for their trip but rather as a complement to other activities.⮚Wine novice. For different reasons (such as advertising along a route or wanting new experiences), these individuals visit wineries without having any knowledge in this field. The main reason for the trip is not associated with wine, but these individuals spend a few hours visiting wineries. The purchases they usually make are intended for private consumption or, in most cases, as gifts.

Each type of enotourist demands a different product [92]. Wine lovers are more demanding in terms of the quality of the wines and the explanations about the production process than are wine novices; they are also more willing to pay more for a quality product.

In Spain, according to the Association of Spanish Wine Cities (ACEVIN) in its 2019 report [93], the number of enotourists who visit PDOs is not homogeneous; 3,076,634 people visited wineries and museums along the wine routes of Spain, more than double that of a decade ago. However, despite being a very significant figure, it is still small compared to the 43 million tourist who visited American wineries in 2017, especially considering that Spain is the leading country in the world in terms of vineyard surface area and the third largest wine producer [94]. One of the most attractive options for enotourists is enjoying the view of a vineyard within a natural setting [95]. Different approaches are being implemented in other countries that may explain these numbers. By region, Andalusia receives the most enotourists. The PDO of Jerez receives more than half a million tourists per year and is the best-known international PDO from the tourist point of view (Figure 4), receiving more than 80% of novice tourists; in the PDO of Ribera del Duero, that same percentage of tourists are wine interested.

Analysis of the profile of enotourism consumers in the Montilla-Moriles (Córdoba) designation of origin: Montilla-Moriles wine spans different municipalities of the province of Córdoba. The main economic activity of the inhabitants of this area is agriculture, followed by the service industry, except in the capital of the province, where the tertiary sector is practically the only sector. Likewise, the secondary sector is practically nonexistent in the area except in the production of wine and oil because there is no other type of manufacturing activity, with approximately 4.5% of jobs being lost in the secondary sector each year.

This area is relatively well connected by road and rail, with the different provincial capitals of its surroundings (mainly Seville, Granada and Malaga). Likewise, it is also close to two international airports, which is a deciding element for citizens of other countries to consume this tourism product.

Wines in the area have recognized prestige thanks to the control exercised by the designation of origin. The quality of the wines in the area is largely due to the clay soil, the climate, the location of the vineyards, the historical legacy of production and the use of new technologies. The wines of this area include fine wines and bitter wines, which are pale gold in color and very aromatic. Regarding vineyards, the “Pedro Ximénez” variety prevails, along with the “Moscatel,” “Lairén,” “Airén,” “Baladí Verdejo” and “Montepila” varieties, grown using tilling, pruning and trellising, culminating with a harvest at the end of August, the earliest in Spain. Grapes are crushed and pressed to extract the must from which the wines will age. After fermentation, the wine is transferred to stacked wooden casks or criaderas to age. Aging takes place in the wineries that dot the periphery of this geographical area, where the ideal temperature, humidity and light conditions are maintained at the levels required for the sophisticated production procedures.

Currently, 60 companies are part of the Regulatory Council of the “Montilla-Moriles” designation of origin, of which 18 are cooperatives (30%), 30 are limited liability companies (50%) and 12 are private companies (20%).

The Montilla-Moriles Wine Route was established at the beginning of 2001 within the framework of the commitments acquired by the City Council of Montilla, as a city of wine, with the Spanish Association of Wine Cities (ACEVIN). In April 2001, the Association for the Promotion of Wine Tourism (AVINTUR) was established and is responsible for man-aging the Montilla-Moriles Wine Route. Currently, 39 public and private entities, in particular wineries and rural accommodations, are part of the route. The route runs through the southern portion of the province of Córdoba and crosses nine municipalities (Aguilar de la Frontera, Córdoba, Fernán Núñez, La Rambla, Lucena Montilla, Montemayor, Moriles and Puente Genil). As shown in Figure 5, the route is composed of three different options. The first (in green) runs through Montilla-Moriles and focuses on the two localities for which the route is named and where the most important wineries are located. The second section (in blue) covers the cities of Córdoba, Fernán-Núñez, Montemayor, La Rambla and Montilla and attempts to unite the important cultural and heritage legacy of the city of Córdoba with rural tourism resources. The third (in pink) covers the cities of Aguilar de la Frontera, Moriles, Lucena and Puente Genil, and the objective of this option is to offer tourists from Costa del Sol an alternative to the traditional sun and beach destinations.

Thus, it is essential to analyze the socioeconomic profile of wine tourists to adapt the supply to the existing demand in each area. The last section of the study presents the conclusions obtained when analyzing the PDO Montilla-Moriles (Córdoba). The results obtained will enable the adaptation of the products offered by small and medium-sized businesses in rural areas to the consumer demand of this tourism segment. Through the diversification of economic activities in rural areas, it will be possible to obtain supplemental income and generate new jobs, alleviating current problems in rural areas.

## 4. Materials and Methods

This investigation focuses on conducting an econometric study to estimate the quantitative demand of wine tourism on the Montilla-Moriles route and to determine the characteristics of tourists who visit this route, with the objective of identifying a tourist profile and, subsequently, the necessary measures to improve this tourist route, which would logically generate an increase in wealth in this area.

The sources of information that have been used to carry out this study are as follows:●Information on the number of monthly tourists who visit route Montilla-Moriles (from January 2015–February 2020).●Data obtained through fieldwork and two different surveys (Table 3):
○The first survey was conducted during the months of February to May 2019 and included companies that are part of AVINTUR and the wineries that belong to the Montilla-Moriles Regulatory Council (in total, 85 businesses); the response rate was 46% (39 surveys received), with a margin of error of 4.7%. The objective of this survey was to determine the enotourism offerings in this area.○The second survey, conducted during the months of February to December 2019, was applied to 500 people who visited this route. To determine the profile of wine tourists, a questionnaire consisting of 35 questions divided into four blocks of tourist consumers who visited the Montilla-Moriles wine route in 2019 was conducted. The first block of the questionnaire collected personal information (e.g., age, gender, educational level, marital status) The second block gathered information about the route taken (e.g., How did you learn about the gastronomic route? Did the route meet your expectations? What would you change? Did you travel expressly because of the gastronomic route?). The third block addressed the motivation for gastronomic tourism (e.g., Why did you choose a gastronomic route?). The fourth block collected information regarding value (e.g., services received on the route, price of the trip, hospitality and treatment received). The objective of this survey was to establish a profile of tourists who choose this type of tourism and to determine the motivation of such tourists, with the purpose of reinforcing and designing strategies that promote the development of wine tourism in the area.

**Table 3 ijerph-19-03393-t003:** Fact sheet for the survey.

	Offer Survey	Demand Survey
Population	Companies that are part of AVINTUR and the wineries that belong to the Montilla-Moriles Regulatory Council	Tourists of any gender over 18 years old who undertook/visited a route/PDO/PGI Montilla-Moriles
Sample size	39	500
Sampling error	±4.7%	±3.9%
Sampling System	Simple Random	Simple Random
Level of confidence	95%; p = q = 0.5	95%; p = q = 0.5
Date of fieldwork	February to May 2019	February to December 2019

Source: Own elaboration.

With the data obtained, the following are proposed: (1) A binary logit model in which the variable under study is dichotomous and to which two values are assigned: 1, which represents the category of the variable to be analyzed, and 0 otherwise. The objective is to determine the probability of tourist satisfaction relative to the expectations they had of the tourist route, based on their socioeconomic profile [96]. (2) A SARIMA model to predict the demand for enotourism in the Montilla-Moriles PDO based on a sample from January 2015 to March 2020. The Box–Jenkins (BJ) methodology was applied using ARIMA models. According to Box et al. (2015) [97], the facilitating factor of this prediction method is an analysis of the probabilistic, or stochastic, properties of economic time series themselves (in this case, the number of wine tourists in the Montilla-Moriles PDO).

Likewise and independent of the logit and SARIMA models, contingency tables were used to establish the relationship between age, income level and the degree of tourist satisfaction. The chi-square test was used to determine associations from contingency tables for the three variables; the value obtained was 75.1, with a probability of 0, indicating that there is a strong association between the three variables studied [98].

## 5. Results

### 5.1. Estimation of the Level of Tourist Satisfaction Relative to Their Expectations, Based on Their Socioeconomic Profile: A Logit Model

In general, the average profile of a tourist who travels the Montilla-Moriles Wine Route is a skilled worker between 50 and 59 years of age who has a medium-high income level, travels as a family, and considers that the treatment received (hospitality) is good but also feels that the cost is high, and that the area lacks complementary activities to the wineries. Some of these characteristics coincide with those noted by Sundbo and Dixit [99], who reported that tourists are predominantly between 40 and 55 years of age and are independent professionals or skilled workers with a medium-high level of training (approximately 50% have studied at university).

In addition, a logit model was estimated based on a sample of 500 people from the European Union; the objective of the model was to calculate the probability of satisfaction relative to tourist expectations regarding the Montilla-Moriles route based on their socioeconomic characteristics. Satisfaction was the variable under study (Satisf), with satisfied taking a value of one and dissatisfied taking a value of zero.

The main predetermined variables in this survey were as follows:-Gender of the respondent;-Age (over 18 years);-Professional activity: acp (professional), ace (employer with workers), acd (manager), acf (civil servant), actc (skilled worker), acta (self-employed worker), aces (student), acam (homemaker), or acj (retired);-Family income (thousands of euros/month): rf;-People with whom the trip was made: s (alone), p (partner), f (family) and a (friends);-Restaurant service: catering (one for good and zero for bad);-Number of times having previously visited the Montilla-Moriles geographic zone: nv;-Expenses incurred during the vacation: gr;-Recommend the geographical area as a tourist destination: re, dichotomous variable (one, yes; zero, no);-Days of vacation used for the type of tourism carried out: dv;-Number of wineries visited: b;-Opinion about lodging (dichotomous variable): oalo (one for good and zero for bad);-Complementary activities: acco (zero, bad; and one, good);-Price of the trip: price (zero, bad; one, good);-Hospitality (zero, bad; one, good);-Conservation of the environment: ce (zero, poor; one, good);-Accommodation (zero, poor; one, good);-Opinion on the information and signage along the route: is (zero, bad; one, good).

The results of the estimation are provided in Table 4:
Satisf = 1/(1 + e^−(β^_0_
^+ β^_1_^gender+ β^_age_^+ β^_3_
^acp + β^_4_^ace + β^_5_^acd + β^_6_
^acf+ β^_7_^actc + β^_8_^acta + β^_9_^aces + β^_10_
^acam + β^_11_^acj + β^_12_^rf + β^_13_^s + β^_14_^p +β^_15_^f + β^_16_^catering + β^_17_
^a + β^_18_^nv +β^_19_^gr + β^_20_^re+ β^_21_
^dv + β^_22_
^b + β^_23_
^oalo + β^_24_^acco+ β^_25_
^price + β^_26_^hospitality +β^_27_^ce + β^_28_^is +)^) + ε(1)

From the estimations, we obtained the following results:■The variable number of wineries visited positively influenced the degree of satisfaction with the trip because as the number of wineries visited increased, the degree of satisfaction was higher (B_22_ = 17.568).■The age variable was also significant because when the age of the tourist increased, their level of satisfaction was higher (B_2_ = 12.253). In our opinion, this result could be related to the profile of wine tourists who visit this area, consistent with the previous classification made by Charters and Ali-Knight [89]. Depending on the profile of tourists, wineries sell different tourism products.■Eighty-four percent of the people surveyed would recommend this tourist route, a result that, in our opinion, reflects the high degree of satisfaction that this destination provides to tourists (B_20_ = 14.572).■Regarding the negative variables indicated by the travelers surveyed, the high price of the trip (B_25_ = −1.253) and the few complementary activities in the area (B_24_ = −4.983) were notable.

The growing importance of wine tourism cannot be denied. The need to propose sustainable models of tourism in areas traditionally dedicated to other economic activities, to prevent errors in the commercialization of tourist spaces, leads to the need to determine exactly what and how tourists want to consume at each specific destination. As such, the main results of our research are as follows:

The number of wineries open to the public on this route is still limited, especially during weekends and long weekends, which implies that the wine supply in the area does not adequately satisfy the actual (and potential) demand, which could be diverted to other oenological destinations. For this reason, it is essential to clearly position this tourist destination to create a brand image for the route, preventing tourists from detouring to other, more or less similar destinations [100].

The demand for enotourism in the area is growing, as seen by the positive coefficient for the trend variable (2325 people). Additionally, there is a high probability that tourists will repeat the experience, thus achieving a high degree of loyalty. This leads us to propose that there is a minimum demand for different companies (especially existing cooperatives) to make investments in this area to satisfy this tourist segment.

### 5.2. Estimation of the Demand for Enotourism in the Montilla-Moriles PDO: SARIMA Model

To predict the demand for enotourism in the Montilla-Moriles PDO, a SARIMA model was used. In Figure 6, a slight increasing trend is observed in the variable demand for enotourism in the years analyzed (January 2015 to February 2020); additionally, this variable has a varying trend, which was corrected with the Box–Cox transformation (λ = 0.4), and average and cycle trends that were corrected through a differentiation in averages and in cycles.

The estimated SARIMA model for forecasting the monthly demand for enotourism is (1,1,1) (0,1,0)^12^.
(1 + 1.078215 B) (1 − B)1 (1 − B^12^) Enotourist^0.4^ = (1 + 0.997914 a_t_
t_ϕ1_ = 26.47281 *   t_θ1_ = −101.7076 *
* Significant parameters for α = 0.05.

As seen in Figure 7, the behavior of the model (thousand tourists) with the estimated data (fitted) is very similar to that of the real data (actual), indicating that the SARIMA model is correct.

Table 5 provides the estimation results for the GARCH model for which the parameters are significant and indicates the absence of conditioned autoregressive heteroscedasticity because the probability of the statistics 0.0109 and 0.02950 is lower than the significance level of 5% (prob. column).
GARCH = 14419635 − 2.393621*a_t−1_^2^

Table 6 provides the predictions obtained with the SARIMA model for the year 2022 and a comparison with the year 2019. The years 2020 and 2021 were omitted because they were atypical years due to the pandemic, during which wineries were closed and individuals were unable to travel within Spain (months of March to June 2020). These predictions are made under the assumption that the pandemic is controlled and the borders are completely open, not with total or partial restrictions against tourist entries.

It is expected that growth will be 3.61% during 2022; that is, 1124 more tourists in 2022 than in 2019 will visit the Montilla-Moriles wine route. However, this figure may increase depending on whether tourists prefer not to travel abroad during the pandemic, but rather prefer to engage in more inland tourism, in particular in rural environments.

If the supply continues to increase, it may be possible to mitigate the seasonality of the demand by advertising the route as an option for residents of other autonomous communities and other countries during their local and autonomous festivals [101,102], with the objective of obtaining a better use of existing resources.

The creation of more complementary activities, such as cultural or gastronomic festivals, should be encouraged because such an approach can attract more mature tourists, thus generating more income for the area but also requiring a greater supply of hotel beds and rural houses.

It is necessary to increase investments in promoting this tourist destination, and illegal offerings must be controlled.

## 6. Discussion

The profile of the gastronomic tourist has changed in recent years, especially as a result of the COVID-19 pandemic. Tourists seek destinations that are safe from a health point of view and sustainable for the environment, preserving cultural heritage and giving back to the local community in terms of the benefits generated by tourism [88,89].

As indicated by Winfree et al. [103], enotourism can help increase wine sales in wineries, especially when tourists classified as wine experts participate; it is also a way of promoting products through tourism. In the case of the Montilla-Moriles route, many wineries, especially those operating as cooperatives, are reluctant to welcome tourism because they think that this activity will not generate an increase in sales, while the individual wineries have a more forward-looking vision and believe that the enotourist is a potential wine seeker in their cities who will contribute to increasing sales and act as an ambassador for their products.

The profile of the tourist along the Montilla-Moriles route is a person of mature age (50 to 59 years), and very similar to that of the studies by Sundbo and Dixit [99], who is approximately 55 years old, but quite far from the enotourist profile of the Jerez-Xérès-Sherry PDO [104] who are mostly young people under 30 years of age. The difference in profiles is because most tourists who visit the Jerez wineries seek sun and beach tourism; their visits to the wineries are a side activity. Wine tourism is not their main purpose of the trip, and this group consists mainly of foreign tourists classified as wine novices. In contrast, enotourists who visit the Montilla-Moriles route can be classified as connoisseurs who know the world of wine and know what wines they can expect to taste on the route. It follows that they are highly satisfied, since they feel that the wines of the Montilla-Moriles PDO are high quality. Many of these tourists have visited other wineries in other areas of Spain, such as La Rioja, which rank higher in terms of wine heritage (wineries, wine museums, signage, etc. [105]. However, we are still a long way from the profile of the tourist who visits the Quinta da Gaivosa wineries in Portugal [106]; these enotourists can be classified in the highest category—wine lovers—indicating that they are a great connoisseur of wine culture, travel expressly for wine-related reasons and earn more than 3000 euros per month, purchasing wine from the winery after the visit. This is the type of profile that the Montilla-Moriles route should be seeking.

The ARIMA model for forecasting wine tourism demand for 2022 in the Montilla-Moriles route predicts that this will grow, but only slightly (3.6%), despite the increase in inland tourism in Spain, especially in rural areas by more than 50% in 2021, as tourists sought nature and to get away from crowded tourist areas. This shows that the Montilla-Moriles area is losing an opportunity to attract new tourists, especially those from the Spanish market, because the route is not well known. Its communication channels should be improved, especially the marketing efforts it has made thus far. Marketing should employ more digital channels to ensure that the ecotourism experience is innovative, multisensory and unforgettable and so that the experience is satisfactory and achieves positive E-WOM [107]. In addition, the wine route must be advertised in specialized wine magazines that will attract wine lovers with high purchasing power. However, if the number of wineries available to be visited on weekends cannot be expanded, then new hotels should be built, the restaurant service improved, and a menu of dishes offered that are either made with local wines or accompanied by local wines. If not, the Montilla-Moriles route will lose an opportunity that will be taken advantage of by other wine tourism routes. This study does not advocate excessively increasing the number of enotourists who would exceed the carrying capacity of the wineries or vineyards but rather that the focus be on improving the quality of tourists to attract people who know how to appreciate the wines of the PDO and are potential buyers of these wines in their home cities. The preferred approach is less tourism but more tourists with higher purchasing power, who will spend the night and whose average daily expenditure is high compared to the current profile of day trippers who spend less than 6 h in the area and who only visit the winery but do not buy the products. The great problem of the province of Córdoba, where the Montilla-Moriles route is located, is that it caters to day excursions, tourists passing through who spend the night in other cities such as Malaga or Seville, leaving behind few economic resources in the region. Enotourism could be a solution to increase provincial wealth, which has one of the lowest average incomes in Spain.

## 7. Conclusions

Tourists seeking nature during the pandemic can help mitigate the socioeconomic gap in rural areas and provide endogenous development. For this, a strategic vision of a sector that integrates agriculture, development and tourism is necessary to prevent saturation of the rural environment and to promote environmental sustainability for generating wealth and employment. This statement coincides with studies from [28,34,35].

An increase in enotourism in the Montilla-Moriles PDO requires the coordination and involvement of all agents, i.e., public and private entities, neighborhood associations, and entrepreneurs, who at all times take into account the quality of the environment; this is the only way to guarantee the continuous offering of tourism products that are the fruits of the effort and tenacity of the people and resources in rural areas.

The Montilla-Moriles area should diversify the tourism products offered on the market and specialize in adapting to the changes in consumer habits and to satisfy their needs, which is ultimately the most important for building loyalty and attracting new customers, in addition to expanding hotel infrastructure and complementary services related to the wine experience.

Obtaining tourist products demanded by consumers is an arduous task; once such products are obtained, they will generate a new source of supplemental income for the inhabitants of the region. Thus, the help of public entities and private entities to enhance the unique elements of the area is mandatory.

In this study, an analysis of the tourist demand of the Montilla-Moriles Wine Route was conducted using different statistical tools. Through the fieldwork carried out, the demand for enotourism and the probability of tourists being satisfied based on their personal characteristics were estimated, identifying the main parameters of actual tourist demand (in short, the client) visiting this area. For both public managers and private enterprises, we propose guidelines regarding the behavior of tourists with respect to medium-term trends.

Based on these conclusions, there may be an opportunity for the development of supplemental economic activities in the Montilla-Moriles area due to the increase in tourism demand (results obtained with the SARIMA model); however, to achieve this development, the support of different public administrators (especially infrastructure improvements) and private companies is necessary. The area should adapt to this new tourist segment, adjusting to the changes in the habits of consumers to satisfy their needs, which, in short, is the most important element for building loyalty and attracting new customers.

An element of debate, or perhaps of reflection, that is open to future research is the different perspectives that tourists and entrepreneurs of the area have regarding what the greatest improvements on this route might be. Most tourists think that there should be more complementary offerings, especially nightlife, to increase the number of overnight stays, and entrepreneurs in the area should focus on promotion and publicity.

This divergent response between the two sides of the market should involve a deep analysis and reflection on the part of the different public administrators involved in the route and the private companies that offer products that effectively cater to the demand of tourists to increase complementary activities and to create an indispensable marketing plan. In turn, these actions should be guided by an in-depth analysis that clearly defines the demand based on the profile of wine tourists who visit this route.

Finally, as a future line of research, we highlight the need to conduct an in-depth study of the relationship that might exist between cultural tourism (focused on the city of Córdoba) and rural tourism (focused on Subbética Natural Park) with this tourist route. Such an investigation could lead to flows of tourists that combine two (or even all three) tourist destinations, therefore resulting in a greater redistribution of tourism-derived income that would be generated in a part of the province of Córdoba similar to that which has occurred in other places [98].

## Figures and Tables

**Figure 1 ijerph-19-03393-f001:**
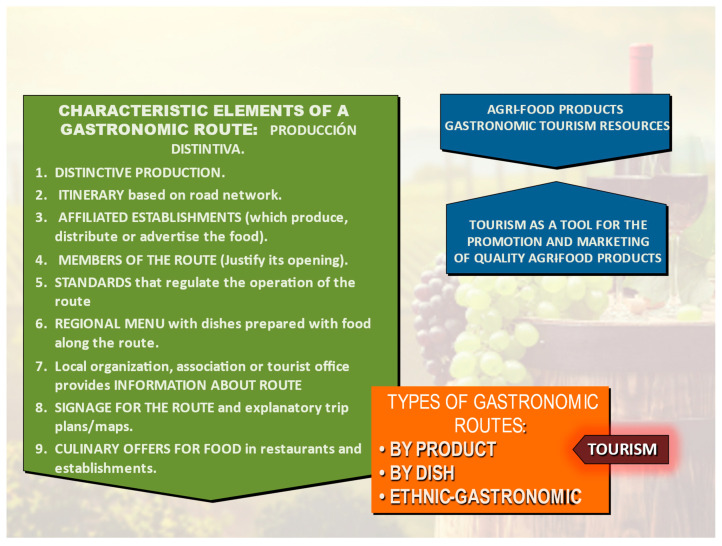
Elements of a gastronomic route. Source: Own elaboration.

**Figure 2 ijerph-19-03393-f002:**
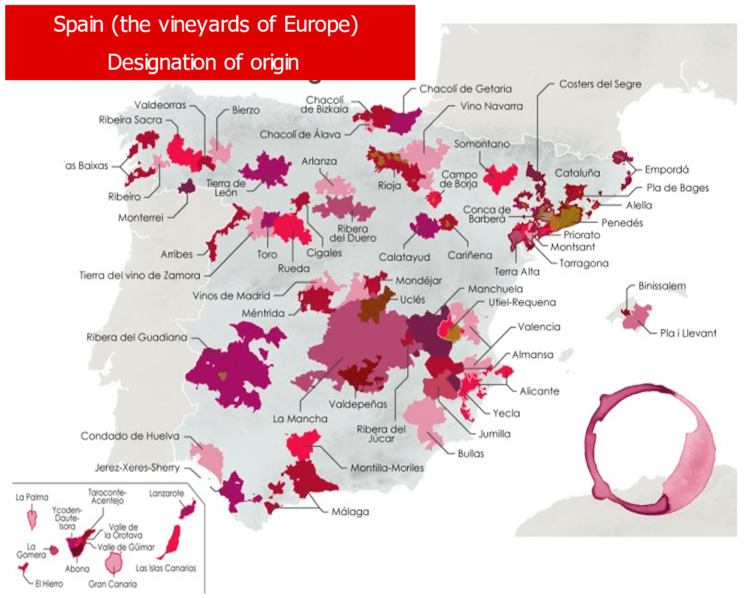
Protected designations of origin for wine in Spain (2021). Source: Ministry of Agriculture, Fisheries and Food (MAPA) [46].

**Figure 3 ijerph-19-03393-f003:**
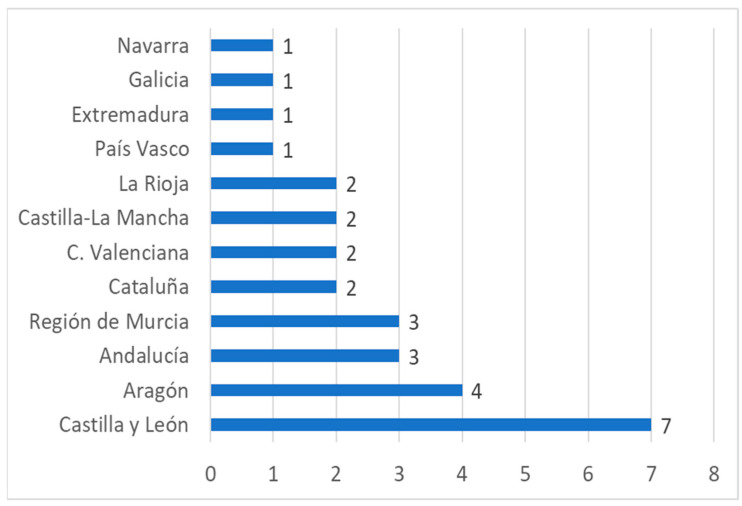
Wine routes in Spain by autonomous community (year 2021). Source: Own elaboration based on data from the Ministry of Agriculture, Fisheries and Food (MAPA) [46].

**Figure 4 ijerph-19-03393-f004:**
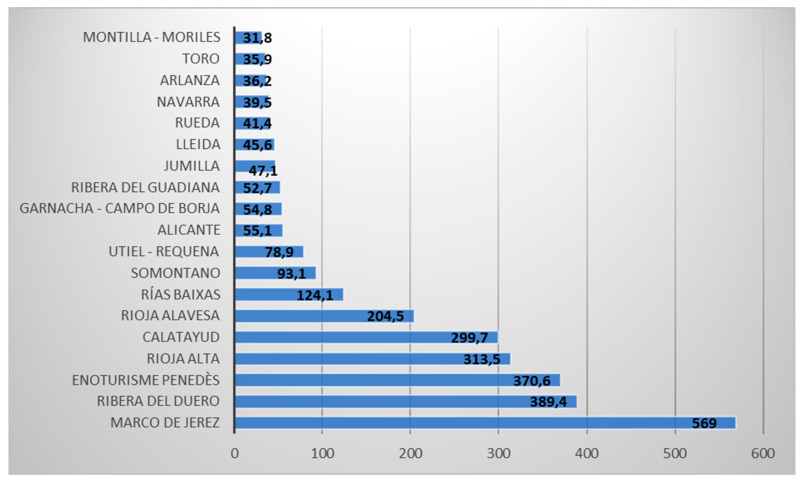
Number of visitors and wine routes in Spain (2019). Source: Own elaboration based on ACEVIN data [93].

**Figure 5 ijerph-19-03393-f005:**
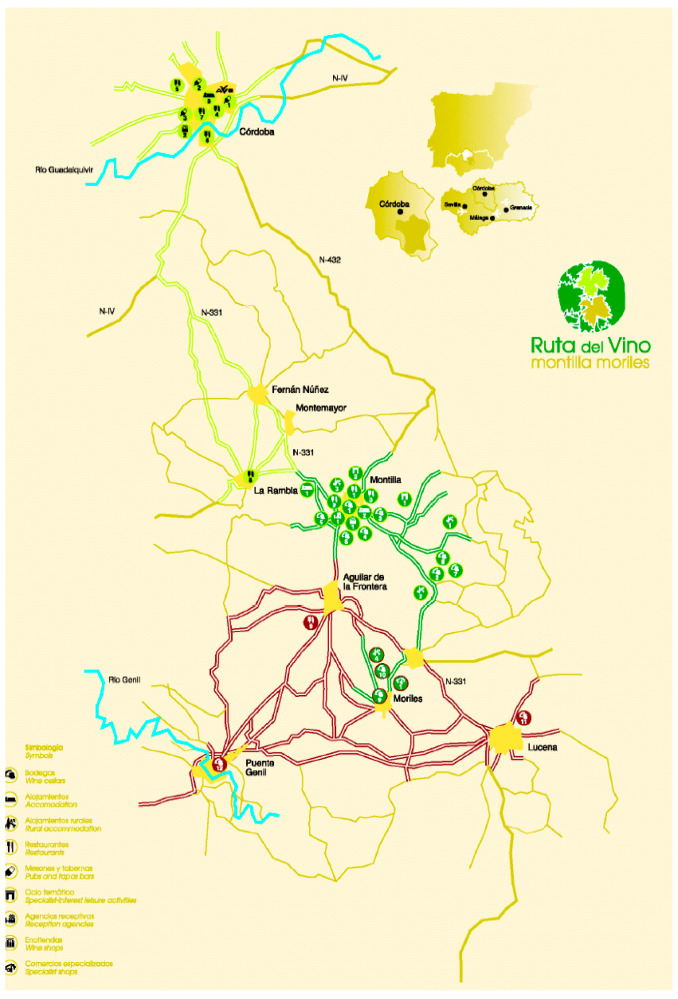
“Montilla-Moriles” wine route. Source ACEVIN.

**Figure 6 ijerph-19-03393-f006:**
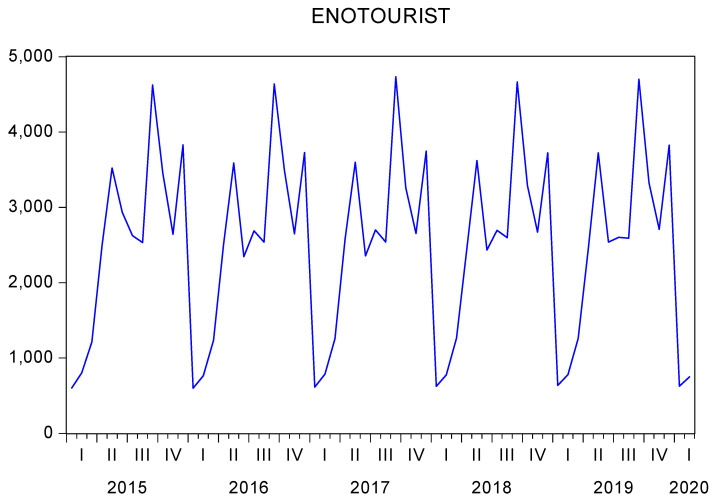
Evolution of enotourism demand in the Montilla-Moriles PDO (thousand tourists, January 2015 February 2020). Source. Own elaboration.

**Figure 7 ijerph-19-03393-f007:**
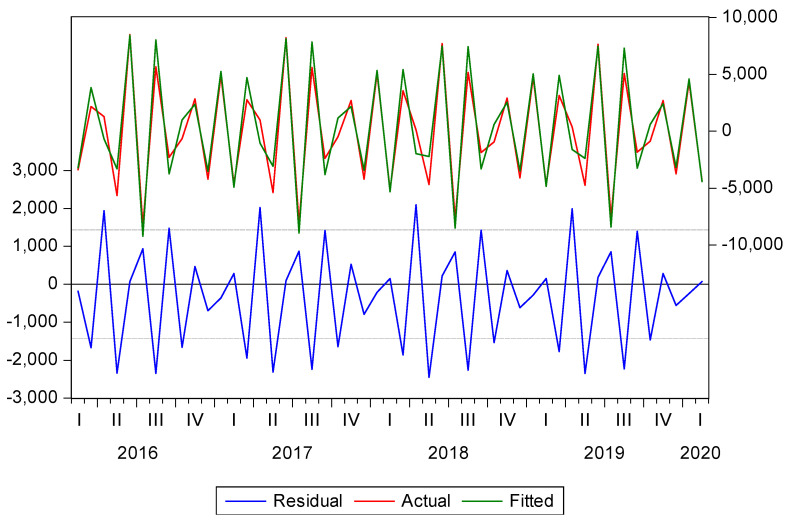
Comparison of real enotourism (actual), estimated enotourism (fitted) and errors (residual). Source. Own elaboration.

**Table 1 ijerph-19-03393-t001:** Protected designations of origin and protected geographical indications of Spain (year 2021).

Agri-Food Products	PDO		PGI	
	Spain	Andalusia	Spain	Andalusia
Fresh meat (and offal)	-	-	22	-
Meat products	5	1	11	2
Cheeses	27	-	2	-
Other products of animal origin (honey	3	1	4	-
Oils and fats (32 oils and 2 butters)	34	13	3	1
Fresh and processed fruit, vegetables and grains	1	-	4	4
Other products (saffron, paprika, chufa (nutsedge), hazelnut, vinegar, cider)	9	3	-	-
Bakery, confectionary, pastry and biscuit products	-	-	16	4
Suckling pig (Cochinilla)	1	-	-	-
**Total PDOs and PGIs for agri-food products**	**106**	**22**	**98**	**13**
Wine with a designation of origin (DO)	73	6	-	-
Wine with a guaranteed designation of origin (DOCa)	2	-	-	-
Wine of quality with a geographical indication (VC)	10	2	-	-
Vinos de Pago (VP) = Quality wines from a single estate that fall outside the DO	11	-	-	-
Wine with a geographical indication (GI)	-	-	42	16
Aromatized wine	-	-	1	1
**Total PDOs and PGIs for WINE**	**96**	**8**	**43**	**17**
Spirits with PGIs	-	-	19	1
**Total PDOs and PGIs**	**202**	**30**	**160**	**31**

Source: Own elaboration based on data from the Ministry of Agriculture, Fisheries and Food (MAPA) [46].

**Table 2 ijerph-19-03393-t002:** Protected designations of origin and protected geographical indications for wine in the European Union (2021).

Country	PDO	PGI
Greece	33	45
Spain	96	43
Italy	420	127
Netherland	9	12
Portugal	36	17
France	478	76
Belgium	8	2
Bulgaria	54	1
Czechia	11	2
Denmark	1	4
Germany	19	26
Cyprus	7	4
Luxembourg	1	
Hungary	49	7
Malt	2	1
Austria	34	3
Romania	42	14
Slovenia	14	3
Slovakia	8	1
**Total**	**1322**	**388**

Source: Own elaboration based on data from the Ministry of Agriculture, Fisheries and Food (MAPA) [46].

**Table 4 ijerph-19-03393-t004:** Logit model estimation.

Dependent Variable: Satisf				
Method: ML—Binary Logit (Quadratic Hill Climbing)				
Variable	Estimated Coefficient	Standard Deviation	Z	Prob
Intercept	B_0_ = 2.325	0.103	22.572	0.010
**Gender**	B_1_ = 0.631	0.002	315.500	0
**Age**	B_2_ = 12.253	2.251	5.443	<0.0002
Professional, **acp**	B_3_ = 9.256	1.425	6.495	<0.0002
Entrepreneur, **ace**	B_4_ = 0.042	0.001	42.000	0
Management, **acd**	B_5_ = 0.035	0.002	17.500	0
Official, **acf**	B_6_ = 13.564	1.561	8.689	<0.0002
Skilled worker, **actc**	B_7_ = 17.891	3.458	5.174	<0.0002
Self-employed, **acta**	B_8_ = 10.584	2.231	4.744	<0.0002
Student, **aces**	B_9_ = −0.058	0.001	−58.000	0
Homemaker, **acam**	B_10_ = 9.856	2.521	3.909	<0.0002
Retired, **acj**	B_11_ = 8.642	1.658	5.212	<0.0002
Family income, **rf**	B_12_ = 15.324	2.567	5.969	<0.0002
Traveling alone, **s**	B_13_ = −0.567	1.261	−0.449	0.3264 *
Traveling as a couple, **p**	B_14_ = 7.368	3.457	2.131	0.0166
Traveling with family, **f**	B_15_ = 14.658	1.578	9.289	<0.0002
**Catering**	B_16_ = 4.328	0.679	6.374	<0.0002
Traveling with friends, **a**	B_17_ = 6.745	2.012	3.352	<0.0002
Number of visits, **nv**	B_18_ = 2.561	0.123	20.821	0
Expenditures, **gr**	B_19_ = 1.568	0.111	14.126	0
Would recommend trip, **re**	B_20_ = 14.572	2.877	5.065	<0.0002
Vacation days, **dv**	B_21_ = 0.045	0.001	45.000	0
Wineries visited, **b**	B_22_ = 17.568	3.684	4.768	<0.0002
Lodging opinion, **oalo**	B_23_ = 0.536	0.014	38.285	0
Complementary activities, **acco**	B_24_ = −4.983	1.021	−4.881	<0.0002
Trip price, **price**	B_25_ = −1.253	0.014	89.500	0
**Hospitality**	B_26_ = 3.762	0.985	3.819	<0.0002
Environmental conservation, **ce**	B_27_ = 1.236	0.021	58.857	0
Information and signage, **is**	B_28_ = 0.023	0.002	11.501	<0.0002

R^2^ McFadden = 0.56. * All the parameters are significant α = 0.05 except B_13_. Source: Own elaboration.

**Table 5 ijerph-19-03393-t005:** Estimation results for the GARCH model.

Dependent Variable: D(ENOTOURIST1,12)				
Method: ML ARCH (Marquardt) Normal Distribution				
GARCH = C(3) + C(4)*RESID(−1)^2^				
Variable	Coefficient	Std. Error	z-Statistic	Prob.
AR(1)	−1.078216	0.040729	−26.47281	0.0000
MA(1)	−0.997914	0.009812	−101.7076	0.0000
	Variance Equation			
C	14419635	5663686.	2.545981	0.0109
RESID(−1)^2^	−2.393621	1.099372	−2.177263	0.0295
R-squared	0.910115	Mean dependent var		−77.28674

Source. Own elaboration.

**Table 6 ijerph-19-03393-t006:** Predictions of enotourism demand for the Montilla-Moriles PDO.

Month	Year 2019	Year 2022	Difference
January	636	735	99
February	779	824	45
March	1258	1328	70
April	2446	2566	120
May	3725	3846	121
June	2538	2624	86
July	2601	2700	99
August	2589	2695	106
September	4701	4928	227
October	3325	3432	107
November	2708	2652	−56
December	3824	3924	100

## Data Availability

The data presented in this study are available on request from the corresponding author.

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
