# Peer review of "Enotourism in Southern Spain: The Montilla-Moriles PDO"

_ijerph, 2022, doi:10.3390/ijerph19063393_

Round 1
Reviewer 1 Report
Congratulations to the Authors of a successful article. I believe they have taken up a very interesting topic. However, the submitted text requires some clarifications and corrections.
1. The introduction lacks a clear emphasis on the purpose of the article. The goal was only presented in the discussion. The introduction is wordy. They need to be shortened - it should be specific.
2. Table 3 is illegible and unnecessarily lengthy. There is no point in presenting the logistic function. Please consider stopping at the table with estimated parameters.
3. The discussion is extremely modest, as the Authors hardly discuss with the previously cited publications. This gives the impression that the authors wanted to limit themselves to conclusions.
I encourage you to correct the article and good luck.
Author Response
Dear Reviewer:
We appreciate your comments and suggestions, and we have made our best efforts to address them.
- The introduction lacks a clear emphasis on the purpose of the article. The goal was only presented in the discussion. The introduction is wordy. They need to be shortened - it should be specific.
We have shortened the introduction, and the objectives of the research have been added to the introduction.
Table 3 is illegible and unnecessarily lengthy. There is no point in presenting the logistic function. Please consider stopping at the table with estimated parameters.
The logit model has been specified outside the table, and the table has been restructured to be more understandable by adding the z statistics and their probability.
The discussion is extremely modest, as the Authors hardly discuss with the previously cited publications. This gives the impression that the authors wanted to limit themselves to conclusions
A discussion section has been added independently of the conclusions, and a new bibliography has been added
Reviewer 2 Report
The paper is very interesting, it deals with interesting and current topics in the world. The authors have very clearly and legibly presented the wine tourism routes of great importance for the area of ​​Spain - Montilla-Moriles Wine Route. The paper is rich in statistical data important for insight into the development of this form of tourism. The tabular graphs are very clear. The authors conducted two surveys, one in May, the other in December 2019. The methodology is completely clear and precise, they used the binary logistics model as well as the SARIMA model. It can be noticed that the results are presented in tabular form, but also very clearly so that readers can find their way in reading the obtained results. The paper has no objections, except that it is emphasized that COVID influenced the change of tourist routes, which should be avoided, because the profile of tourists has changed over the years, which was not fully influenced by the pandemic. I would like to praise the authors for their research work and the way they presented the research to us readers. I suggest that the paper be published without changes.
Author Response
Dear Reviewer:
The paper is very interesting, it deals with interesting and current topics in the world. The authors have very clearly and legibly presented the wine tourism routes of great importance for the area of ​​Spain - Montilla-Moriles Wine Route. The paper is rich in statistical data important for insight into the development of this form of tourism. The tabular graphs are very clear. The authors conducted two surveys, one in May, the other in December 2019. The methodology is completely clear and precise, they used the binary logistics model as well as the SARIMA model. It can be noticed that the results are presented in tabular form, but also very clearly so that readers can find their way in reading the obtained results. The paper has no objections, except that it is emphasized that COVID influenced the change of tourist routes, which should be avoided, because the profile of tourists has changed over the years, which was not fully influenced by the pandemic. I would like to praise the authors for their research work and the way they presented the research to us readers. I suggest that the paper be published without changes.
We appreciate your comments. Although you indicated publishing the work directly, we have made some small changes that we believe will improve the article, such as separating the discussion from the conclusions

Reviewer 3 Report
Dear Author(s),
Thank you very much for this interesting article.
This research, according to the authors “focuses on conducting an econometric study to estimate the quantitative demand of wine tourism on the Montilla-Moriles route and to determine the characteristics of tourists who visit this route, with the objective of identifying a tourist profile and, subsequently, arbitrate the necessary measures to improve this tourist route, which would logically generate an increase in wealth in this geographical area”. I applaud the paper's involvement in touristic discussions.
The abstract needs, in my opinion, a complete rethinking. I couldn’t read, in this section, the main arguments, methodology, the most important aims and the critical contribution of the paper to the economical dimension of wine tourism.
I recommend to the author(s) to reinforce the theoretical background of this research. In my view, it could be interesting, for example, to adding a ‘literature review’ section to organise the current version of the paper directly related to its theoretical contribution.
There are some suggestions to reinforce the literature review of this research:
Cristófol FJ, Cruz-Ruiz E, Zamarreño-Aramendia G. Transmission of Place Branding Values through Experiential Events: Wine BC Case Study. Sustainability. 2021; 13(6):3002. https://doi.org/10.3390/su13063002
Cruz-Ruiz E, Zamarreño-Aramendia G, Ruiz-Romero de la Cruz E. Key Elements for the Design of a Wine Route. The Case of La Axarquía in Málaga (Spain). Sustainability. 2020; 12(21):9242. https://doi.org/10.3390/su12219242
Güzel, Ö., Ehtiyar, R., & Ryan, C. (2021). The Success Factors of wine tourism entrepreneurship for rural area: A thematic biographical narrative analysis in Turkey. Journal of Rural Studies, 84, 230-239.
Nave, A., Laurett, R., & do Paço, A. (2021). Relation between antecedents, barriers and consequences of sustainable practices in the wine tourism sector. Journal of Destination Marketing & Management, 20.
Winfree, J., McIntosh, C., & Nadreau, T. (2018). An economic model of wineries and enotourism. Wine Economics and Policy, 7(2), 88-93.
This article, in general terms, and according to my view, is too descriptive. I perfectly understand that is a study case (The Montilla-Moriles PDO), but I have serious doubts on the contribution of this article to a high impact journal as International Journal of Environmental Research and Public Health.
A critical issue of this article is, in my opinion, the difference between “to comment a case study” or “to analyse a case study”. The current version of the paper offers data and contents that are very close to ‘comment a case study’. For this reason, I firmly recommend some changes to the author(s):
1. Reinforce the discussion and concluding remarks section. It is important to create a valuable debate on the basis of another case studies, and according to the most recent scientific literature that has been published on the relationship between destination marketing, wine tourism and economic development (I suggested some references on the previous sections of this review).
2. It is necessary to create a strong and clear link between scientific literature and the case study. This is the only way to advance in the knowledge that now exists in the intersection between local development and tourism. I will be very happy if I have the chance to read more arguments and reflections on this issue in the discussion and conclusions of this paper.
3. Consider how you present your descriptive data related to the case study analysis, in order to be clearer in differentiating between data and analysis, primary and secondary data, and to draw out and emphasize your main findings and make sure that these relate to issues and questions identified in the early parts of the paper.
4. This is a very interesting contribution for two different audiences: scholars and practitioners. Please, rewrite your proposal on the basis of both profiles.
For all of these reasons, and in my opinion, the background data of the case is essentially informative, and also, I think that the case study developed in the paper has an important potential and offers a valuable data. According to this, I really think that the current version of the paper needs to clarify its critical contribution to the relationship between economic development and wine tourism.
I hope that the author(s) finds the energy to apply these suggested major changes to the current version of the paper and do not find my criticism too hard. Good luck.
Author Response
Dear Reviewer:
We appreciate your comments and suggestions, and we have made our best efforts to address them all.
The abstract needs, in my opinion, a complete rethinking. I couldn’t read, in this section, the main arguments, methodology, the most important aims and the critical contribution of the paper to the economical dimension of wine tourism.
The abstract has been modified by adding the demand forecasting model.
I recommend to the author(s) to reinforce the theoretical background of this research. In my view, it could be interesting, for example, to adding a ‘literature review’ section to organise the current version of the paper directly related to its theoretical contribution.
We thank the reviewer for the suggestion of a new bibliography that has been added and has also served to improve the discussion. An additional bibliography has also been added.
There are some suggestions to reinforce the literature review of this research:
A critical issue of this article is, in my opinion, the difference between “to comment a case study” or “to analyse a case study”. The current version of the paper offers data and contents that are very close to ‘comment a case study’. For this reason, I firmly recommend some changes to the author(s):
- Reinforce the discussion and concluding remarks section. It is important to create a valuable debate on the basis of another case studies, and according to the most recent scientific literature that has been published on the relationship between destination marketing, wine tourism and economic development (I suggested some references on the previous sections of this review).
A discussion section has been added independent of the conclusions, and a new bibliography has been added.
- It is necessary to create a strong and clear link between scientific literature and the case study. This is the only way to advance in the knowledge that now exists in the intersection between local development and tourism. I will be very happy if I have the chance to read more arguments and reflections on this issue in the discussion and conclusions of this paper.
We have reinforced the discussion section. It now appears as a section independent of the conclusions.
- Consider how you present your descriptive data related to the case study analysis, in order to be clearer in differentiating between data and analysis, primary and secondary data, and to draw out and emphasize your main findings and make sure that these relate to issues and questions identified in the early parts of the paper.
We have attempted to answer the questions posed in the paper using the results of the logit and SARIMA models.
- This is a very interesting contribution for two different audiences: scholars and practitioners. Please, rewrite your proposal on the basis of both profiles.
We have made the requested changes and believe the article is now clearer

Round 2
Reviewer 3 Report
Dear Author(s),
Thank you for submitting the revised version of your paper. I'm very happy to announce you that, in my opinion, the paper has been considerably improved.